# Contextualizing Evidence for Action on Diabetes in Low-Resource Settings—Project CEAD Part I: A Mixed-Methods Study Protocol

**DOI:** 10.3390/ijerph17020569

**Published:** 2020-01-16

**Authors:** Elisa Chilet-Rosell, Nora Piay, Ildefonso Hernández-Aguado, Blanca Lumbreras, Francisco Barrera-Guarderas, Ana Lucía Torres-Castillo, Cintia Caicedo-Montaño, Gregorio Montalvo-Villacis, Mar Blasco-Blasco, María Fernanda Rivadeneira, María Pastor-Valero, Mónica Márquez-Figueroa, Juan Francisco Vásconez, Lucy Anne Parker

**Affiliations:** 1CIBER de Epidemiología y Salud Pública (CIBERESP), 28029 Madrid, Spain; 2Department of Public Health, Universidad Miguel Hernández, 03550 Sant Joan d’Alacant, Alicante, Spain; 3Faculty of Medicine, Pontificia Universidad Católica del Ecuador (PUCE), Quito 170143, Ecuador; 4Institute of Public Health, Pontificia Universidad Católica del Ecuador (PUCE), Quito 170143, Ecuador; 5Centre of Community Epidemiology and Tropical Medicine (CECOMET), Esmeraldas 0801265, Ecuador; 6School of Medical Specialities, Colegio de Ciencias de la Salud, Universidad San Francisco de Quito (USFQ), Quito 170901, Ecuador; 7Faculty of Nursing, Pontificia Universidad Católica del Ecuador (PUCE), Quito 170143, Ecuador; jfvasconezd@hotmail.com

**Keywords:** diabetes mellitus, type 2 diabetes, primary prevention, public policy, public health, implementation science

## Abstract

Challenges remain for policy adoption and implementation to tackle the unprecedented and relentless increase in obesity, diabetes and other non-communicable diseases (NCDs), especially in low- and middle-income countries. The aim of this mixed-methods study is to analyse the contextual relevance and applicability to low-resource settings of a sample of evidence-based healthy public policies, using local knowledge, perceptions and pertinent epidemiological data. Firstly, we will identify and prioritise policies that have the potential to reduce the burden of diabetes in low-resource settings with a scoping review and modified Delphi method. In parallel, we will undertake two cross-sectional population surveys on diabetes risk and morbidity in two low-resource settings in Ecuador. Patients, community members, health workers and policy makers will analyse the contextual relevance and applicability of the policy actions and discuss their potential for the reduction in inequities in diabetes risk and morbidity in their population. This study tackles one of the greatest challenges in global health today: how to drive the implementation of population-wide preventative measures to fight NCDs in low resource settings. The findings will demonstrate how local knowledge, perceptions and pertinent epidemiological data can be used to analyse the contextual relevance and applicability of potential policy actions.

## 1. Introduction

Diabetes mellitus is a global health emergency: it affects all age groups in all countries across the globe [1], being especially prevalent in low and middle-income countries (LMIC) where 75% of people with diabetes live [2]. Type II diabetes accounts for more than 90% of the cases worldwide [3] and its risk factors include not only individual and behavioural factors, but also structural ones. Structural factors determining the rise in diabetes include rapid unplanned urbanization together with globalization, which in turn lead to the greater accessibility of alcohol, tobacco and unhealthy foods, and reduced physical activity. Furthermore, poor governance exacerbates inequalities in terms of distribution of power, money and resources, which ultimately means that large swathes of people remain in poverty and at risk of malnutrition and infectious diseases, even though obesity and non-communicable diseases (NCDs) like diabetes are climbing [4,5]. 

The International Diabetes Federation estimates that 5.5% (3.5–8.5) of Ecuador’s population aged 20–79 years suffers from diabetes, of whom 40% are undiagnosed [2]. In 2016, diabetes was the second leading cause of mortality among women and third in men [6]. Considering that Ecuador is among the most rapidly aging countries in South America, and that 62% of the population aged 20–59 are overweight or obese [6], it is likely that diabetes will continue to be a significant health problem for the country. Globally, it is estimated that the number of people affected by diabetes will rise to 642 million in 2040, being most pronounced in LMIC [7]. The increasing interest in a reduction in the risk of diabetes led to the development and implementation of population-wide initiatives in LMIC, i.e., India [8,9]. 

Given these dramatic global projections [3,7], most of the major international organizations and policy-making bodies have called for healthy public policies. The World Health Organization (WHO) recommends a combination of fiscal policies, legislation, changes to the environment to promote healthier diets and physical activity, and raising awareness of health risks. Solutions for the prevention and control of NCDs require a whole systems approach that integrates action on the social determinants of health with the consideration that individual choices are constrained by social, environmental, economic, political and cultural factors [10].

Observational studies that explore the nature of existing inequalities, and studies that describe experiences of tackling upstream determinants of health (for example, by placing restrictions on advertising, or introducing fiscal measures on unhealthy products) have an important role in generating evidence for policy action to fight NCDs globally, but implementation remains limited in many parts of the world. Implementation in LMICs may be especially challenging, given that the evidence base for action on NCDs comes primarily from high income countries. Innovative ways to drive the implementation of population-wide preventative measures are desperately needed to fight NCDs globally. The process of knowledge translation requires evidence being weighted up, interpreted, assessed for relevance, applied to the here and now of a particular local setting and considered in light of competing priorities and concerns [11]. Furthermore, implementation research is needed to understand and address the barriers to the acceptance and adoption of NCD strategies [12]. 

This current protocol is part of a five year (2019–2023) European Research Council (ERC, Brussels, Belgium)-funded research project which aims to explore the process by which global recommendations can be translated into context-specific, evidence-informed action for diabetes prevention in low-resource settings. “Contextualizing Evidence for Action in Diabetes in Low-resource settings”, acronym CEAD, is an implementation science project and, as such, we do not propose to analyse the impact of public health interventions, but to analyse how contextual factors may affect their relevance, and to explore the processes involved in their adoption and implementation. The project design combines quantitative epidemiological research and qualitative methodologies, involving an intricate feedback loop to generate the rich and varied knowledge that is required to trigger policy action [13]. 

This current report describes the protocol for the first part of the project, which deals with the relevance and implementation of healthy public policies at local or municipal levels. 

## 2. Materials and Methods 

The aim of this study is to analyse contextual relevance and applicability to low-resource settings of a sample of evidence-based healthy public policies, using local knowledge, perceptions and pertinent epidemiological data.

### 2.1. Study Design

Mixed methods study: firstly, we will identify the sample of evidence-based policies that have the potential to reduce the burden of diabetes, given the available evidence, using a scoping review. In parallel, we will undertake two cross-sectional population surveys, with representative samples of two selected health districts (an urban and a rural district) to estimate the prevalence of diabetes, impaired glucose tolerance, and NCD risk factors locally. The knowledge generated from the scoping review and the population surveys will be discussed in contextualization discussion groups, where participants will discuss the contextual relevance and applicability of the policy actions identified and their ability to reduce the health inequities observed in their district (Figure 1). 

### 2.2. Identification of a Set of Policy Actions

#### 2.2.1. Scoping Review

The aim of the scoping review is to map the evidence regarding policies that are recommended at local or municipal level to promote healthy diets and increase physical activity in different contexts in both high-income countries and LMICs. Only policy actions that are focussed on modifying the environment rather than addressing individual behaviour modifications will be considered. We will follow Arksey and O’Malley’s six-stage methodological framework [14] and the guidance to standardize the conduct and reporting of the scoping reviews developed by the Joanna Briggs Institute [15]. 

##### Search Strategy

This scoping review will consider guidelines and recommendations from principal international health agencies. First, we will identify public health agencies or organizations that generate evidence-based recommendations and are supported by governmental departments, but which operate independently, such as the Community Preventive Services Task Force of the USA and National Institute for Health and Care Excellence of the UK, and the WHO and regional offices. Agencies or institutions will be included if they have an explicit description of procedures to produce evidence-based recommendations. We will then systematically search the recommendations made by each agency/organization to extract the information related to policy actions. The search strategy will go on until: (a) no more agencies/institutions that provide evidence-based recommendations and are supported by governmental departments or multilateral institutions are found; or (b) saturation of information regarding policy recommendations is reached.

##### Extraction of Results

Two reviewers will extract data related to the policy actions independently and any disagreements will be resolved through discussion and consensus, or by a third reviewer in cases of disagreement. The following information will be extracted: the guidelines or documents that support the action; year of publication; general recommendations (e.g., promoting active travel to school or meal or fruit and vegetable snack school interventions); specific actions (e.g., building bicycle pathways or pricing healthier foods and beverages); intervention source (organisation); target environment (e.g., schools, worksites or community); target population (e.g., youth, elderly, people who are overweight or obese); and the intended audience of the recommendation (e.g., local policy makers, health managers), according to the powers and responsibilities of the local governments. 

#### 2.2.2. Delphi Method

We will use an adapted Delphi method with experts and members of the research team to prioritise a set of policy actions, taking into consideration their applicability in low resource settings and, more specifically, to the health districts included in the study. The Delphi technique has emerged, since its inception, as one of the most powerful and frequently used formal consensus techniques [16]. We will adapt the Delphi technique by including a first phase where participants will make independent ratings privately, using a Likert scale of 1 to 5, regarding the technical applicability of the political action in a series of settings (low resource settings, rural areas, urban areas), and their opinion of the cost-effectiveness of the action in these settings. In this phase, the participants will be provided with the opportunity to make qualitative comments to justify and/or qualify their ratings. In a second phase, all the information gathered in the first phase will be shared with the participants and they will be asked to rank all potential policies according to their perceived appropriateness (considering technical applicability and cost-effectiveness) for (a) low-resource urban settings and (b) low-resource rural settings. The next round will be repeated iteratively until consensus is achieved on approximately 10 preferred policies for each setting. 

### 2.3. Population Surveys

#### 2.3.1. Setting

We will carry out the fieldwork for this case study in two carefully selected, low-resource settings in Ecuador: (1) District 17D06, Quito, an urban health district with 507,499 residents (est. 2017); and (2) District 08D02, in Esmeraldas Province, a rural area in the northern coastal region of the country with a population of 44,498 (est. 2017). The decision was made in part due to the higher diabetes prevalence in the capital and coastal areas (observed in the 2012 national survey for health and nutrition [6]) and, most importantly, due to the presence of local researchers working in the health arena, motivated to address the challenges of implementing evidence-based diabetes care in the population and collaborate with the study team.

#### 2.3.2. Study Participants

Residents aged over 18 years who provide informed consent will be eligible for inclusion in the population surveys. We define a resident as an individual who was sleeping at least 20 days of the previous month in a residence in the district and who has no plan to move in the near future. Individuals who are unable to provide informed consent (e.g., significant mental impairment) will be excluded. 

#### 2.3.3. Sample Size

A sample of 720 people per district will lead to a total sample of 1440 individuals for the population surveys. We propose this sample size assuming the prevalence of diabetes is no higher than 10%, to provide an estimate with an absolute precision of ±3%, with a 95% confidence level. We assume a design effect of 1.5 (recommended in the WHO STEPS guideline for complex designs in absence of a locally available alternative [17]) and further increase the sample size to allow for up to 20% refusal or loss due to individuals not attending their appointment for physical and biological measurements. 

#### 2.3.4. Sampling Strategy

We will use multi-stage cluster sampling. Sampling in urban settings will consist of a random selection of neighbourhoods (primary enumeration unit), whose probability of being chosen will be proportional to the population density. Once we have selected the neighbourhood, we will identify buildings or homes by randomly generated GPS points and will randomly sample one person at each point. In rural settings, we will conduct a multistage stratified cluster sampling. Villages in the rural settings of Esmeraldas are distributed along the Santiago River, where the Cayapas River and Onzole River also converge. In line with the existing geographic and demographic organization, we will randomly select ‘micro-areas’, defined as groups of villages pertaining to a single primary health facility. We will then select a random sample of villages stratified by ethnicity (indigenous, afro-Ecuadorian, mixed) with disproportionate allocation, and by isolation (distance from the main urban area following the riverbeds). Finally, we will undertake simple random sampling of eligible individuals from within the villages by using an existing census developed by voluntary health promotors, digitised by the Centre of Community Epidemiology and Tropical Medicine (CECOMET), collaborators in the project.

#### 2.3.5. Measurements and Procedures

The questionnaire used in the population survey will include sections of the WHO STEPS NCD risk factor survey forms [17]. Detailed demographic information will also be included following the WHO STEPS core questions and extended questions (highest level of education, ethnicity, marital status, employment status and household income), with cultural adaptations to the questions as required. 

Twenty-four interviewers will establish survey eligibility, solicit informed consent, and carry out the survey in the participant’s home. The composition of the survey team will be gender balanced, and include staff who represent local cultural, ethnic and religious groups. They will carry out the survey in Spanish but some interviewers will also be fluent in Quechua, Chachi and any other relevant indigenous language to translate questions verbally if required. Data collection in rural areas will engage the volunteer health promotors under the supervision of CECOMET research staff. The surveyors will arrange an appointment for physical and biological measurements together with the participant, considering geographical convenience and availability. 

Weight, height, waist circumference and blood pressure will be measured according to the technical specifications recommended in the STEPS guideline [17] in a separate area to ensure participant privacy. A blood draw will be carried out in a well-lit hygienic location after at least 8 h fasting. Glucose tolerance will be measured with an Oral Glucose Tolerance Test (OGTT) requiring a second blood draw 2 h later. Participants will receive instructions for fasting on the appointment card on the survey day and telephone reminders will be sent when possible. Diabetic patients on medication will be asked to bring their medication and take it immediately after their breakfast after the blood draw. Diabetic patients and pregnant women are eligible to participate in the study but they will not undergo an OGTT. A nationally accredited laboratory will analyse plasma glucose, cholesterol and creatinine from venous blood samples. We will take steps to minimise variability (within each of the two study sites) in the pre-analytical treatment of samples. Blood extraction, storage and transportation will follow standard infection control measures.

All participants will receive the results of their physical measurements and blood analyses and information about diabetes, cardiovascular risk, renal function and the importance of a healthy diet and physical activity. Information regarding adverse effects associated with incidental findings in otherwise healthy populations will be shared with the survey staff in order to avoid over-diagnosis and over-medicalization. Only individuals with abnormal glucose levels, blood pressure, renal function or hypercholesterolemia according to the current National Clinical Practice Guidelines will be referred to their local health facility for confirmation and consequent care as per the national protocols.

#### 2.3.6. Data Collection and Analysis

For data collection in the survey, we will use the WHO STEPS forms where possible and import them to Kobo toolbox (http://www.kobotoolbox.org/) free open source software (or ODK collect if necessary). We will add prompts for immediate error checking of unlikely values. We will pre-pilot the survey tool before the survey, after training the data collection team. We will use stickers with Quick Response (QR) codes that can be scanned by the tablets in order to ensure the linkage of data collected in participant’s home, their physical and biological measurements and laboratory results. 

We will use Stata/SE (StataCorp, College Station, TX, USA) Version 15 for statistical analysis. We will calculate the prevalence of impaired glucose tolerance, diabetes, and the other NCD risk factors with 95% confidence intervals. We will disaggregate outcomes by sex, age group, ethnicity and socioeconomic position. We will also describe geographical variation in survey outcomes. Steps will be taken to prevent missing data, but some level of missing data is unavoidable and we will incorporate methods analysing missing data or data from uncertain sources when necessary [18,19].

### 2.4. Contextualization Discussion Groups

We will organise 10 contextualization discussion groups, five in each setting: Quito and Esmeraldas. Discussion will be based on information obtained from two different sources: (1) a sample of approximately 10 potential policy actions obtained from the scoping review and modified Delphi procedure (ensuring actions to promote both physical activity and healthy diet are included) that can be implemented at a local level; (2) the results obtained from the population survey showing the prevalence and distribution of diabetes, hypertension, obesity and other NCD risk factors.

#### 2.4.1. Participants and Sampling Strategy

Contextualization discussion groups will include 4–8 participants and last approximately 90 min. Participants must be older than 18, residents in the study area, and they must provide written informed consent. The sampling method used for the selection of contextualization discussion groups participants will be purposive and will include diabetic patients and family members, community members, healthcare workers and decision-makers (Figure 2). In each setting, there will be two discussion groups of participants A and B, two discussion groups with participants B and C, and a final group including all types of participant (A, B, C and D) and may also include local food producers and distributers if avoiding undue influence in the discussion is deemed feasible. 

#### 2.4.2. Procedures

Discussions will be guided by the information obtained from policy actions prioritised using the Delphi methodology previously described and the results of the population survey of the prevalence and distribution of diabetes, hypertension, obesity and other risk factors for non-communicable diseases, focusing on health inequalities. 

We will present all information in a visual and simplified format. Participants will receive the written information and a short explanation at the beginning of the activity and they will have approximately 10 min to read it and ask clarification questions. At this point, researchers will inform participants that they will be audio recorded in order to have all the information. Participants are expected not only to talk about their personal experiences, ideas and opinions, but also to listen to and respect other members of the contextualization discussion groups. 

We will divide discussions into two parts. During the first part, we will ask participants about the feasibility and applicability of the selected potential actions in their context. During the second part of the discussion, facilitators will display the survey results. Participants are expected to talk about the differences observed between population groups (health inequality), suggest potential reasons for the observed inequalities and suggest solutions. Finally, participants will analyse the policy actions one by one and discuss the capacity of the health policies to tackle health inequity in their context. 

#### 2.4.3. Analysis

At least two members of the research team will be present at the discussions; one will facilitate the discussion and the other will have an observatory role, quietly noting observations related to body language, timepoints and his/her perception of the power dynamic. All group discussions will be audio recorded and transcribed. We will analyse digitised transcripts using thematic analysis, assisted by Atlas.ti 8.0 (Scientific Software Development GmbH, Berlin, Germany). 

## 3. Ethics Approval and Consent to Participate

The study protocol has been reviewed and approved by the Universidad Miguel Hernández (UMH) project evaluation board (registration number 2018.291.E.OEP), the nationally accredited ethical board at the Pontificia Universidad Católica de Ecuador (PUCE, reference 2019-27-MB), and ethical clearance has been provided by the European Research Council Executive Agency (ERCEA, Ref. Ares (2018) 5827042-14/11/2018). An independent Ethics Adviser has been appointed and we will monitor ethical and participatory issues, paying particular attention to gender and other equity concerns throughout the research. 

Furthermore, written authorization to conduct the study and review clinical records in public health facilities will be obtained from the National Health Intelligence Directorate of the Ministry before any data collection in Ecuador takes place. 

## 4. Informed Consent 

Informed consent is required prior to participation in the population survey and the discussion groups. On all forms, it is clearly stated that the participant is free to withdraw from the study at any time for any reason without prejudice to future care, and with no obligation to give a reason for withdrawal. Specific arrangements will be taken in order to avoid any inequity derived from any social condition, giving the participants as much time as needed to consider the information, and the opportunity to question the researcher, their health care provider or other independent parties before they decide whether they will participate in the study. Written informed consent will be obtained by means of the participant’s dated signature and the dated signature of the person who presented and obtained the informed consent. 

Although participants will be able to withdraw consent at any time, a special explanation will be given to participants of the qualitative components of the research. Specifically, while focus group participants may choose to leave the debate at any time, after the group discussion is completed and thematic analysis has been carried out we will be unable to erase their input because no personal data will have been collected. 

## 5. Discussion

The current proposal is a timely one. The political basis for concerted action to address diabetes is integrated into the Sustainable Development Goals [20], the United Nations Political Declaration on the Prevention and Control of NCDs [21] and the WHO NCD Global Action Plan [11], and tackling diabetes is currently perceived as a health priority in many countries, including Ecuador. Data collected by volunteer community health promotors in one of the survey settings included here show that, among 331 known diabetics living in the community, there were four diabetes-related deaths and at least 34 registered cases of severe complications in 2016 alone (including 19 cases of diabetic foot, three of which resulted in amputation, and three cases of kidney failure; *CECOMET*, *unpublished data*). While existing activities in the area focusing on patient education are clearly important, they are not sufficient. This study aims to analyse population-wide preventative measures (policies) to create health-promoting environments that help individuals make healthy choices and avoid death or disability due to diabetes. Furthermore, by applying the equity lens we will consider whether such population-wide initiatives to promote physical activity and a healthy diet are able to reach the people who need them most. 

Given the nature of the research question, patient and public involvement is essential to the analysis and interpretation of key findings. A description of their selection and role is described above. Furthermore, two project advisory committees consisting of health workers, patients and community members will be established at the start of the study (one in Esmeraldas, one in Quito). The committees will represent the interests of study participants, meeting regularly throughout the study period to obtain information on the concerns of patients, health workers and the community. Community engagement activities will involve specific events in health centres and other community meeting-points to share the findings of the research, giving members of the community the opportunity to question or comment. The findings will be shared in a non-paternalistic manner, integrated with culturally appropriate entertainment and healthy refreshments.

Regarding knowledge dissemination, we will encourage interaction with stakeholders through sharing and discussing findings at professional meetings (e.g., county health meetings) and academic events at participating universities. This will be a two-way process that will encourage the generation of contextually tailored evidence for the Ministry of Public Health in Ecuador, as well as generalised tools that can help contextualize local policy options in other low-resource settings. We will also share key messages drawn from the research on the study website (under development). 

We will use social media to encourage the circulation, interpretation and contestation of knowledge in both professional and lay networks. Finally, aside from community engagement in Quito and Esmeraldas, we will organize an event in Spain coinciding with World Diabetes Day (14 November 2020) where stakeholders will have the opportunity to share their opinions and reflect on the relevance of the findings in other low-resource settings. Results will also be presented in scientific conferences and seminars, and will be written up for publication in peer-reviewed journals.

The datasets generated by the project CEAD will be archived in the data repository Zenodo, and will be linked in the study website (under construction at time of writing). 

Zenodo will assign a Digital Object Identifier (DOI) to make the upload easily and uniquely citable. Zenodo will allow us to store our (meta) data safely and will allow us to establish different licenses and access levels depending on the type of data (for example, raw data will be only available after the publication of papers).

The DOI of (meta) data will be included in the relevant publications to find the raw (but de-identified) data easily. Raw data will be freely available after the publication of research articles, including the transcripts of the interviews and focus groups. Furthermore, we will make the full research protocols, standard operating procedures, and analysis plans publicly available.

## Figures and Tables

**Figure 1 ijerph-17-00569-f001:**
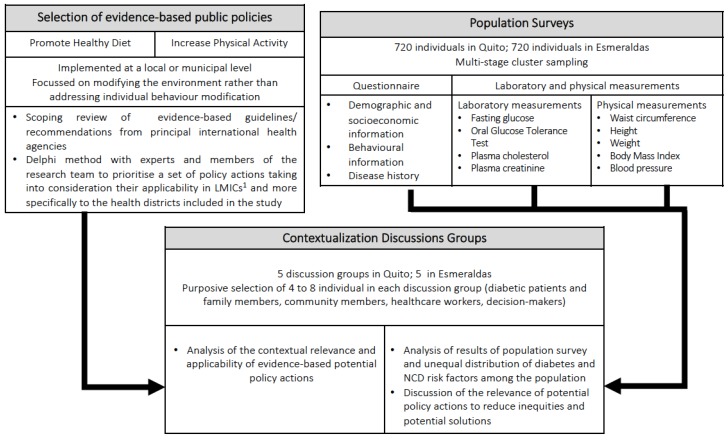
Study design used to analyse the contextual relevance and applicability to low-resource settings of a sample of evidence-based healthy public policies, using local knowledge, perceptions and pertinent epidemiological data. ^1^ LMICs: Low- and middle-income countries.

**Figure 2 ijerph-17-00569-f002:**
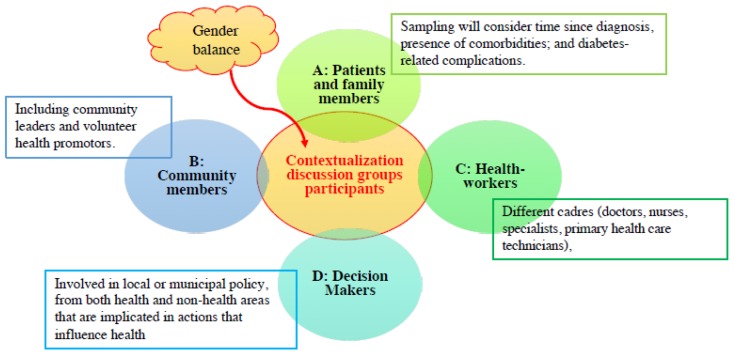
Purposive sampling of participants for contextualization discussion groups.

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
