# Peer review of "Contextualizing Evidence for Action on Diabetes in Low-Resource Settings—Project CEAD Part I: A Mixed-Methods Study Protocol"

_ijerph, 2020, doi:10.3390/ijerph17020569_

Round 1
Reviewer 1 Report
Chilet-Rosell and colleagues wrote this method paper describing the protocol of the project "Contextualizing Evidence for Action in Diabetes in Low-resource settings" - CEAD. With the present paper authors also aim to describe contextual relevance and applicability to low-resource settings of such initiatives. The paper is well-written and relevant as it provides a thoughtful insight on how to implement similar initiative in low-resource settings facing an increasing burden of diabetes. Please find below some comments authors might consider to improve the manuscript:
In the study design section authors state that data collected will be used to estimate IGT. This is very important considering the increasing focus on intermediate hyperglycaemia worldwide. However, this could be covered in the introduction as well - considering that even middle income countries such as India have considered implemented population-wide initiatives targeting this high-risk population In study participant section authors state that will include everybody who was sleeping for at least 20 days in the area and has no intention to leave. This seems quite vague and relaxed. Could you please justify reasons for this choice? Have authors considered that given the use of the OGTT, which can be considered a laborious test, the % of drop could be higher than using non-fasting (but less reliable) blood glucose tests? There is no mention about the statistical approach that will be adopted to analyse collected data. Although the purpose of this paper is different, a mention to that might be useful anywayAuthor Response
Dear Reviewer 1,
Thank you for your kind comments to our paper. Below, we provide a point by point response to your comments and suggestions.
In the study design section authors state that data collected will be used to estimate IGT. This is very important considering the increasing focus on intermediate hyperglycaemia worldwide. However, this could be covered in the introduction as well - considering that even middle income countries such as India have considered implemented population-wide initiatives targeting this high-risk population
Thank you for your suggestion. We have included this information in page 2, lines 55-57 with 2 bibliographic references: "The increasing interest on the reduction of the risk of diabetes lead to the development and implementation of population wide initiatives in LMIC.[8, 9]"
In study participant section authors state that will include everybody who was sleeping for at least 20 days in the area and has no intention to leave. This seems quite vague and relaxed. Could you please justify reasons for this choice?
The reason to state the need to include those who slept for at least 20 days in the area is to include in our study rural migrant population. This population can be in the city on working days (from Monday to Friday) and move to their place of origin on weekends (Saturday and Sunday). If they spend from Monday to Friday in the city, they would spend 20 days7nonth in the city, enough time to consider the context influence in their health.
Have authors considered that given the use of the OGTT, which can be considered a laborious test, the % of drop could be higher than using non-fasting (but less reliable) blood glucose tests?
Although more resource intensive, laboratory glucose measurements were chosen over capillary blood glucose because the latter will likely underestimate the prevalence of diabetes, and will therefore generate less accurate estimates. Glucose tolerance will be measure using Oral Glucose Tolerance Tests (OGTT), as fasting Plasma Glucose test alone fails to identify nearly 30% of cases.
We calculated our sample size considering 20% of loss (see page 6, lines 175-180)
There is no mention about the statistical approach that will be adopted to analyse collected data. Although the purpose of this paper is different, a mention to that might be useful anyway
We included a brief description in page 6, lines 232-237.
Reviewer 2 Report
Please see the following suggested edits:
Line 48--Suggestion--The International Diabetes Federation estimates that 5.5%......
Line 49--In 2016, diabetes was 49 the second leading cause of mortality among women and third in men.
Line 52-- overweight or obese [6], it is likely that……
Line 77-78--….we do not propose to analyse the impact of public health interventions….
Author Response
Thank you so mucha for your suggestions. We have included all of them